# Adoption of HIV pre-exposure prophylaxis among women at high risk of HIV infection in Kenya

**Cedric H. Bien-Gund**[1,2]*, **Perez Ochwal**[3], **Noora Marcus**[4], **Elizabeth F. Bair**[4], **Sue Napierala**[5], **Suzanne Maman**[6], **Kawango Agot**[3], **Harsha Thirumurthy**[4]

**1** Division of Infectious Diseases, Department of Medicine, University of Pennsylvania Perelman School of Medicine, Philadelphia, PA, United States of America, **2** Department of Biostatistics, Epidemiology, and Informatics, University of Pennsylvania, PA, United States of America, **3** Impact Research and Development Organisation, Kisumu, Kenya, **4** Division of Health Policy, University of Pennsylvania Perelman School of Medicine, Philadelphia, PA, United States of America, **5** Women's Global Health Imperative, RTI International, Berkeley, CA, United States of America, **6** Department of Health Behavior, Gillings School of Global Public Health, University of North Carolina at Chapel Hill, Chapel Hill, NC, United States of America

* Cedric.bien-gund@pennmedicine.upenn.edu

**Data Availability Statement:** ***PA/E AT ACCEPT*** Request URL/DOI "Yes - all data are fully available without restriction; All data files will

## Abstract

In 2017, Kenya became one of the first African countries to provide pre-exposure prophylaxis (PrEP) in its national HIV prevention plan. We sought to characterize factors associated with PrEP uptake and persistence among a cohort of women at risk of HIV infection during the early stages of PrEP scale-up in Kenya. HIV-negative women ≥18 years with ≥2 sexual partners in the past 4 weeks were recruited as part of an ongoing cluster randomized trial of an HIV self-testing intervention. PrEP use was assessed at baseline and at 6- and 12-month follow-up visits. Between June 2017 and August 2018, 2,086 were enrolled and had complete baseline data. 138 (6.6%) reported PrEP use during the first year of the study. Although PrEP use increased, persistence on PrEP was low, and less than half of individuals reported continuing PrEP at follow-up visits. In multivariate analyses, PrEP use was associated with recent STIs, having an HIV-positive primary partner, having regular transactional sex in the past 12 months, and being a female sex worker. In the early stages of PrEP scale-up in Kenya, uptake increased modestly among women with risk factors for HIV infection, but overall uptake and persistence was low.

## Introduction

Young women on the African continent account for a quarter of new HIV infections worldwide [1], and nearly three-quarters of all new HIV infections in sub-Saharan Africa [2]. Among female sex workers (FSW), prevalence estimates exceed 50% in some countries in sub-Saharan Africa [3]. In the counties surrounding Lake Victoria in Kenya, HIV prevalence in the general population reaches nearly 20 percent [4] and there is therefore a vital need for effective HIV prevention services.

be available after acceptance of the manuscript for publication in a public data repository.

**Funding:** This research was supported by grants from the National Institutes of Health, R01MH111602-05 (Dr. Thirumurthy) and T32AI055435-16A1 (Dr. Bien-Gund). The funders had no role in study design, data collection and analysis, decision to publish, or preparation of the manuscript.

**Competing interests:** The authors have declared that no competing interests exist.

Widespread implementation of daily oral pre-exposure prophylaxis (PrEP) has shown considerable promise as a way to reduce transmission of HIV [5]. As PrEP programming has been brought to scale, a significant focus of programs has been to increase PrEP utilization and persistence among women at risk of HIV infection in sub-Saharan Africa. In May 2017, Kenya became one of the first African countries to offer PrEP as part of its national plan to end the HIV epidemic [6,7]. Data on PrEP uptake and retention in large-scale implementation efforts have only recently begun to emerge, and factors that influence PrEP uptake and retention remain poorly understood. In two demonstration projects integrating PrEP into family planning and in maternal and child health clinics in Kenya, roughly 20% of women screened and counseled for PrEP initiated PrEP and about 50% had at least one medication refill visit [8,9].

As scale-up of PrEP has increased in Kenya and other countries, there are limited data from population-based studies on the characteristics of individuals who have initiated PrEP [10]. A recent review of PrEP uptake in Africa found that most data have come from implementation programs and demonstration projects [11], which may not reflect the efforts of routine public sector programming. Ascertaining factors associated with PrEP use is useful for implementation of PrEP programming to determine whether PrEP is reaching individuals at high risk, specifically young women with risk factors for HIV infection. Therefore, we sought to characterize PrEP uptake, persistence, and predictors of PrEP uptake among a cohort of women at high risk of HIV infection, during the two years immediately after PrEP scale-up began in public sector health facilities in Kenya.

## Methods

### Study setting, design, and participants

We analyzed PrEP uptake among a prospective cohort of women participating in a cluster randomized controlled trial of an HIV self-testing intervention in Siaya County, Kenya (ClinicalTrials.gov NCT03135067). Between June 2017 and August 2018, women were recruited from beach communities and "hotspots" where transactional sex was common. Enrollment in the study happened to begin one month after PrEP scale-up was initiated in Kenya. During the early roll-out in Kenya, PrEP delivery efforts were focused on areas with high HIV incidence and offered in HIV clinics, antenatal clinics, and drop-in centers for key populations, including for sex workers and adolescent girls and women [7,9]. Indications for PrEP include confirmed HIV-negative status, and any of the following: a sexual partner known to be HIV positive and without an undetectable viral load, a sexual partner of unknown HIV status and at risk for HIV infection, engaging in transactional sex, history of a sexually transmitted infection, use of post-exposure prophylaxis, inconsistent condom use, and sero-discordant couples trying to conceive [12].

For inclusion in this study, eligibility criteria included confirmed HIV-negative status, age ≥18 years, and reporting ≥2 sexual partners in the past 4 weeks. Individuals were excluded if they were enrolled in another HIV prevention research study.

Study clusters were matched based on population and cluster type (beach community or hotspot) and randomized 1:1 to two study groups. Women in intervention clusters received multiple HIV self-tests for distribution to sexual partners while women in control clusters received multiple referral vouchers for clinic-based testing. Study procedures are described in greater detailed elsewhere [13].

### Procedures

Trained study staff conducted a baseline questionnaire and follow-up questionnaires at 6-monthly intervals in Kiswahili. Baseline questionnaires asked about participant's socio-

economic and demographic characteristics, sexual behavior, HIV testing history, and intimate partner violence (IPV). PrEP use was assessed at all study visits. Study staff did not actively encourage or refer participants to seek PrEP at health facilities.

## Measures

The primary outcome of interest was PrEP uptake, which was determined on the basis of participants' report of current PrEP use at any of the study visits (baseline, 6, and 12 months). PrEP uptake was determined to have occurred if participants who reported yes to the question: "*Are you currently taking any HIV medication in order to prevent acquiring HIV (PrEP)?*" Participants who reported PrEP use at any of the study visits were considered to have met the primary outcome. PrEP persistence was defined as self-reporting continuation of PrEP use at a follow-up visit if participants reported PrEP use at the previous visit. Participant characteristics were assessed at baseline, including participant age, education, relationship status, primary income source, and monthly income. HIV and STI measures included self-reported STI diagnosis in the past 6 months and most recent HIV testing. Sexual risk behavior variables included having a primary partner, partner HIV status, condom use with primary partner, number of non-primary partners in the past month, condom use with non-primary partners, and regular or repeated transactional sex over the past 12 months. Study participants also reported on any experiences of IPV (including physical, psychological, and sexual) and pregnancy in the past 6 months.

## Statistical analysis

Socio-demographic characteristics and risk behavior information was measured descriptively. We summarized PrEP use at baseline, 6-month, and 12-month follow-up visits. In order to examine factors associated with PrEP use, we used generalized estimating equations to account for correlation between participants within the study sites. Multivariable models were adjusted for age (continuous), monthly income (continuous, converted to US dollars), and education (categorical), and were determined a priori by the study group. Separate multivariable models were built for each exposure and the primary outcome combination. All data analyses were completed using Stata 15.1 (StataCorp, College Station, TX).

## Ethics statement

All participants provided witnessed signed informed consent to participate in the study. The study received approval from the Institutional Review Boards at the University of Pennsylvania and Maseno University.

## Results

Between June 2017 and August 2018, 2,102 women were initially enrolled in the study. Sixteen women were excluded from the analyses, resulting in 2,086 women being included in the analyses. Ten were withdrawn because they had double-enrolled, 4 women had missing baseline data, and 2 were subsequently determined to be ineligible based on HIV test results from dried blood spots collected at baseline.

A total of 138 (6.6%) women reported PrEP uptake during the first year of the study. At baseline, 1.7% (35/2,086) reported PrEP uptake. At the 6-month visit, an additional 46 women reported PrEP initiation, and at the 12-month visit, an additional 57 women reported new PrEP initiation (Fig 1). PrEP persistence was low. Among the 35 women who reported PrEP uptake at baseline, 15 (42.9%) and 8 (22.9%) reported persistence on PrEP at 6- and 12-month

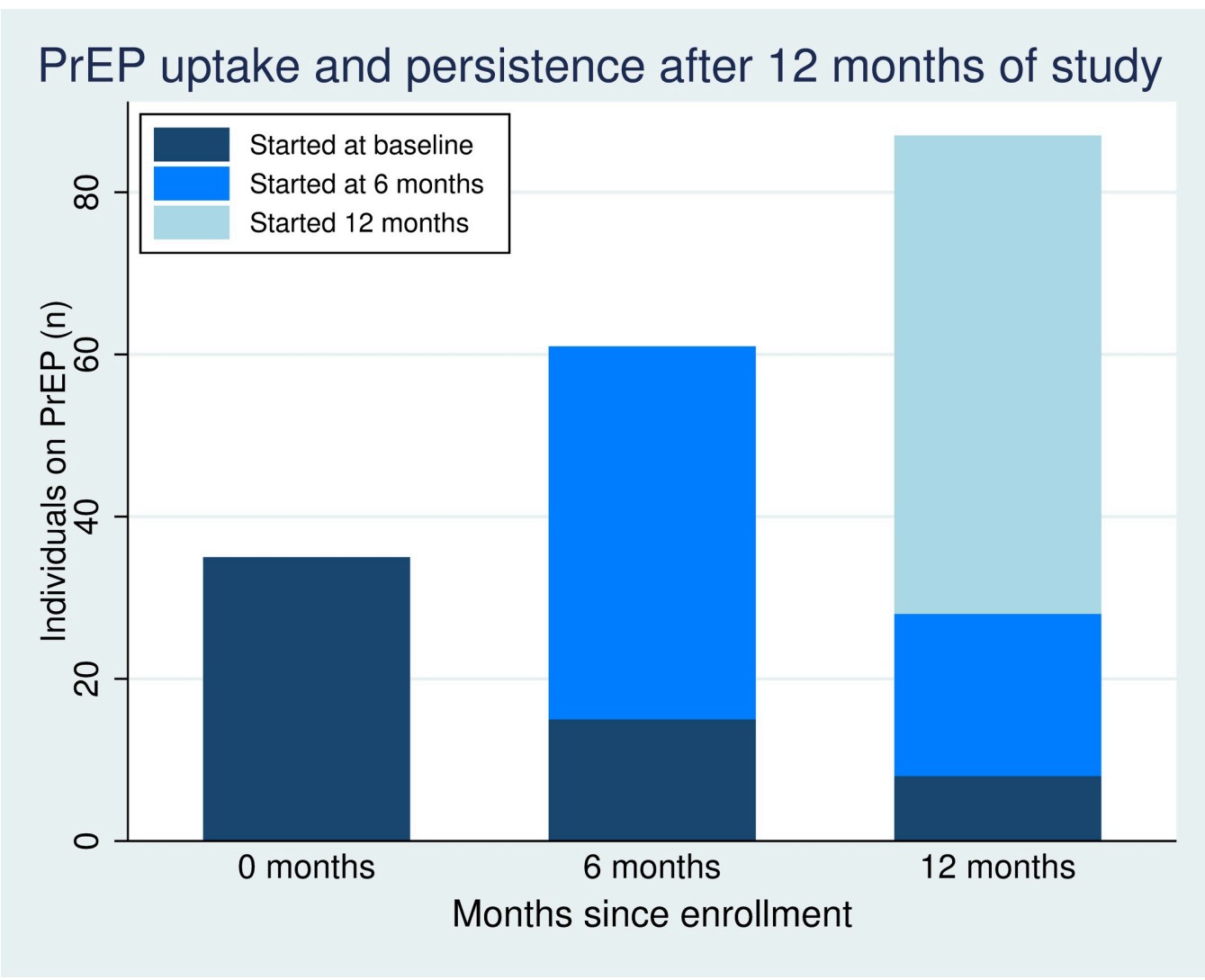

**Fig 1. PrEP uptake and persistence at baseline, 6 months, and 12 months after study enrollment.**

follow-up, respectively. Among the 61 women who reported PrEP at 6-month follow-up, 28 (45.9%) reported PrEP use at 12-month follow-up. We did not observe an association between PrEP uptake and study arm (P = 0.65).

Among women who reported PrEP at any time during the 12 months of observation, the median age was 25 (Table 1). More than half had completed primary school (60.2%), were married (58.0%), and the mean monthly income was 29 USD. Most had a primary partner (90.6%). 26.1% reported sex work as their primary source of income, 85.1% reported regular or repeated transactional sex in the past 12 months. 1.2% of participants reported an STI in the preceding 6 months.

In the multivariable models (Table 2), PrEP uptake was associated with not being in a relationship (AOR 1.60, 95% CI 1.07–2.40), primary income from sex work, (AOR 1.91, 95% CI 1.25–2.93), having an HIV positive primary partner (AOR 5.01, 95% CI 2.30–10.94), and having at least two non-primary partners in the past month (AOR 1.57, 95% CI 1.09–2.25), having regular or repeated transactional sex over the past 12 months (AOR 2.06, 95% CI 1.08–3.95),

**Table 1. Baseline characteristics of HIV-negative women in Siaya County (N = 2,086).**

| Characteristic | No PrEP uptake | PrEP uptake | p-value |
|---|---|---|---|
| | N = 1,948 (%) | N = 138 (%) | |
| Age, median (IQR) | 25.0 (22.0–31.0) | 25.0 (23.0–31.0) | 0.19 |
| Education | | | 0.016 |
| Less than complete primary | 610 (31.3%) | 55 (39.9%) | |
| Complete primary or some secondary | 952 (48.9%) | 68 (49.3%) | |
| Complete secondary or higher | 386 (19.8%) | 15 (10.9%) | |
| Relationship status | | | 0.011 |
| Married | 1,265 (64.9%) | 80 (58.0%) | |
| In a relationship, not married | 190 (9.8%) | 8 (5.8%) | |
| No Relationship | 493 (25.3%) | 50 (36.2%) | |
| Female Sex Worker | 286 (14.7%) | 36 (26.1%) | <0.001 |
| Monthly income in USD, median (IQR) | 29.4 (19.6–58.8) | 29.4 (19.6–49.0) | 0.66 |
| Has a primary partner | 1,872 (96.1%) | 125 (90.6%) | 0.002 |
| Partner HIV status | | | |
| Negative | 999 (53.4%) | 59 (47.2%) | <0.001 |
| Positive | 29 (1.5%) | 10 (8.0%) | |
| Unknown/Won't Disclose | 844 (45.1%) | 56 (44.8%) | |
| Condom use with non-primary partners | | | |
| Never | 551 (42.1%) | 37 (38.9%) | 0.77 |
| Less than to more than half the time | 255 (19.5%) | 21 (22.1%) | |
| Always | 503 (38.4%) | 37 (38.9%) | |
| Non-primary partners in the past month | | | |
| 0 | 2 (0.1%) | 0 (0.0%) | 0.075 |
| 1 | 1,016 (52.2%) | 57 (41.3%) | |
| Any contraception (other than condoms) in the past month | 1,161 (59.6%) | 100 (72.5%) | 0.003 |
| Regular or repeated transactional sex in the past 12 months | 1,563 (84.6%) | 124 (92.5%) | 0.013 |
| STI diagnosis in the past 6 months | 19 (1.0%) | 6 (4.3%) | 0.005 |
| Any intimate partner violence (IPV) | 993 (51.0%) | 59 (42.8%) | 0.062 |
| IPV—sexual | 240 (12.3%) | 14 (10.1%) | 0.45 |
| Pregnant in the past 6 months | 324 (20.6%) | 20 (16.5%) | 0.29 |

and a self-reported STI diagnosis in the past 6 months (AOR 4.49, 95% CI 1.71–11.82). Women who initiated PrEP were less likely to report having a primary partner (AOR 0.48, 95% CI 0.25–0.95). We did not observe a statistically significant association with condom use with non-primary partners, IPV, pregnancy, or time since last HIV test.

## Discussion

During the early stages of the national PrEP rollout in Kenya, we observed low rates of overall PrEP uptake in a cohort of women at high risk of HIV infection, with less than 7% of women reporting PrEP use during the study. Persistence on PrEP was low and declined by more than half at 6-month follow-up visits. PrEP uptake was strongly correlated with recent self-reported STI diagnoses, having multiple partners, having an HIV-positive partner, as well as engaging in transactional sex.

The overall rate of PrEP uptake we observed was lower than previously published literature from PrEP implementation programs for women in African countries. Several implementation programs in Kenya have reported roughly 20% of women screened started PrEP [8,9,14],

**Table 2. Factors associated with PrEP uptake among HIV-negative women.**

| Factor | Unadjusted OR | 95% CI | p-value | Adjusted OR | 95% CI | p-value |
|---|---|---|---|---|---|---|
| Relationship status | | | 0.09 | | | 0.03 |
| Married | Ref | | | Ref | | |
| Relationship, not married | 0.65 | 0.32, 1.33 | | 0.71 | 0.33, 1.53 | |
| No Relationship | 1.35 | 0.94, 1.94 | | 1.60 | 1.07, 2.40 | |
| Primary income from sex work | 1.73 | 1.18, 2.53 | 0.005 | 1.91 | 1.25, 2.93 | 0.003 |
| Has a primary partner | 0.52 | 0.29, 0.94 | 0.03 | 0.48 | 0.25, 0.95 | 0.02 |
| HIV positive primary partner | 4.31 | 2.35, 7.89 | | 5.01 | 2.30, 10.94 | <0.001 |
| Condom use with non-primary partners | | | 0.74 | | | 0.59 |
| Never | Ref | | | Ref | | |
| Less than to more than half the time | 1.22 | 0.73, 2.02 | | 1.32 | 0.76, 2.28 | |
| Always | 1.04 | 0.67, 1.63 | | 1.07 | 0.66, 1.73 | |
| Non-primary partners in past month | | | | | | |
| 0–1 | Ref | | | Ref | | |
| 2 or more | 1.52 | 1.09, 2.13 | 0.01 | 1.57 | 1.09, 2.25 | 0.01 |
| Regular or repeated transactional sex in the past 12 months | 2.02 | 1.08, 3.78 | 0.02 | 2.06 | 1.08, 3.95 | 0.03 |
| Last HIV test | | | 0.29 | | | 0.22 |
| <3 months | Ref | | | Ref | | |
| 3–12 months | 0.79 | 0.55, 1.12 | | 0.76 | 0.52, 1.11 | |
| 13 months or more | 0.66 | 0.32, 1.35 | | 0.60 | 0.28, 1.29 | |
| STI diagnosis in the past 6 months | 3.50 | 1.66, 7.37 | 0.001 | 4.49 | 1.71, 11.82 | 0.002 |
| Any intimate partner violence (IPV) | 0.81 | 0.59, 1.12 | 0.21 | 0.77 | 0.54, 1.09 | 0.14 |
| IPV—sexual | 0.87 | 0.52, 1.46 | 0.59 | 0.93 | 0.52, 1.55 | 0.67 |
| Pregnant in the past 6 months | 0.79 | 0.50, 1.26 | 0.32 | 0.79 | 0.50, 1.25 | 0.31 |

Separate multivariable models for each factor were constructed adjusting for age, education, and income.

however, overall uptake in our study was less than 7%. Higher levels of PrEP initiation have been observed in demonstration projects focusing on FSWs in Zimbabwe [15], Benin [16], and Kenya [17]. The lower uptake in our study than in demonstration projects may be because there was less active promotion of PrEP for women participating in our study, compared to women who were reached by demonstration projects in which they were actively offered PrEP. PrEP availability through the public sector may have also been limited for study participants. Barriers to PrEP uptake include low risk perception of HIV, concern of male partners, and stigma associated with HIV and PrEP [5,9,18]. In addition, the additional clinical resources and training associated with PrEP prescribing to an overburdened public health system may have diminished the initial impact of the PrEP roll-out [19]. We did observe increasing rates of PrEP use at 6- and 12-month follow-up, suggesting that national uptake of PrEP is increasing and ongoing. These findings support the continued integration of PrEP into existing health systems and increasing PrEP awareness and demand [5].

Consistent with previously published data, we observed low persistence on PrEP, with less than 50% of women reporting persistence on PrEP at 6-month follow-up visits [8,14,20]. Sub-optimal persistence on PrEP is a pervasive challenge to successful PrEP implementation. In an implementation study of PrEP among pregnant and post-partum women, less than 40% continued PrEP after the first month, and having an HIV-positive partner was the only predictor of continuation of PrEP [9]. Although risk profiles among women may have changed during

the study, changes in risk perception of HIV, pill burden, lack of social support, and stigma have been described as barriers to persistence on PrEP among women in Kenya [8,17]. From a PrEP provider perspective, reduced barriers to PrEP prescribing, including reducing the frequency of clinic visits and longer refill dates, as well as sending clinic reminders may result in higher rates of persistence on PrEP [19]. Interventions to support adherence and persistence on PrEP may borrow approaches from ART adherence strategies, such as simplified PrEP delivery models (including pharmacy-deliveredwhen or other non-clinical sites), adherence clubs, and stigma reduction activities [11,19].

Our finding that PrEP use was associated with engaging in transactional sex, having a recent STI diagnosis, and having an HIV positive partner are consistent with the Kenya national guidelines for PrEP use [21]. National implementation of PrEP may have the greatest impact among individuals with the highest risk for HIV acquisition, however, we observed gaps in PrEP uptake among certain risk factors. For instance, despite IPV screening in PrEP guidelines and the high prevalence of IPV in our cohort, we did not observe an association between PrEP use and IPV. Universal access to PrEP and more inclusive risk assessments may be needed to optimize PrEP uptake.

This study has several limitations. PrEP use was assessed by self-report at baseline and follow-up questionnaires. Prior clinical research of PrEP uptake among women in sub-Saharan Africa suggest there may be social desirability bias in study settings among women who take PrEP. Nevertheless, because this study was not conducted to evaluate PrEP, we did not expect women to over-report PrEP use. On the other hand, women may have under-reported PrEP and sexual risk behaviors due to stigma associated with HIV and with taking PrEP. Second, we did not collect clinical or biomarker data on PrEP adherence, a key component to efficacy, and did not examine factors associated with persistence due to the small number of women reporting PrEP persistence. Additional studies are needed to evaluate PrEP adherence among women in sub-Saharan Africa. Finally, we did not collect data on where women obtained PrEP, which may have impacted their uptake and persistence.

In summary, PrEP use is increasing during scale-up and appears to be reaching women at high risk of HIV acquisition in Kenya. However, given low rates of PrEP uptake among this cohort of women, there is significant room for PrEP expansion and improving low rates of PrEP persistence. Future research can elucidate barriers and facilitators to PrEP use, adherence, and persistence among women at risk of HIV infection. Development, implementation, and evaluation of PrEP delivery models are needed to further expand PrEP use and persistence.

## Supporting information

**S1 File. Baseline survey (English).**
(DOCX)

**S2 File. Baseline survey (Dholuo).**
(DOCX)

**S3 File. Baseline survey (Kiswahili).**
(DOCX)

**S4 File. 6-month follow-up (Duoluo, English).**
(DOC)

**S5 File. 6-month follow-up (Kiswahili, English).**
(DOC)

**S6 File. Dataset for study: Baseline_6mo_12mo_22Aug2022.**
(DTA)

## Author Contributions

**Conceptualization:** Cedric H. Bien-Gund, Harsha Thirumurthy.

**Data curation:** Cedric H. Bien-Gund, Noora Marcus, Elizabeth F. Bair.

**Formal analysis:** Cedric H. Bien-Gund, Noora Marcus, Elizabeth F. Bair, Sue Napierala, Harsha Thirumurthy.

**Funding acquisition:** Harsha Thirumurthy.

**Investigation:** Sue Napierala, Suzanne Maman, Kawango Agot, Harsha Thirumurthy.

**Methodology:** Elizabeth F. Bair, Harsha Thirumurthy.

**Project administration:** Perez Ochwal, Noora Marcus, Elizabeth F. Bair, Sue Napierala, Suzanne Maman, Harsha Thirumurthy.

**Resources:** Noora Marcus, Elizabeth F. Bair, Sue Napierala, Suzanne Maman, Kawango Agot, Harsha Thirumurthy.

**Software:** Elizabeth F. Bair.

**Supervision:** Harsha Thirumurthy.

**Writing – original draft:** Cedric H. Bien-Gund.

**Writing – review & editing:** Cedric H. Bien-Gund, Perez Ochwal, Noora Marcus, Elizabeth F. Bair, Sue Napierala, Suzanne Maman, Kawango Agot, Harsha Thirumurthy.

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
