## [Decision Letter · Decision Letter 0]

20 Aug 2021

PONE-D-21-19374

Adoption of HIV pre-exposure prophylaxis among women at high risk of HIV infection in Kenya

PLOS ONE

Dear Dr. Bien-Gund,

Thank you for submitting your manuscript to PLOS ONE. After careful consideration, we feel that it has merit but does not fully meet PLOS ONE’s publication criteria as it currently stands. Therefore, we invite you to submit a revised version of the manuscript that addresses the points raised during the review process.

We look forward to receiving your revised manuscript.

Kind regards,

Joseph K.B. Matovu, Ph.D.

Academic Editor

PLOS ONE

Journal Requirements:

4. Thank you for stating the following in the Acknowledgments/Funding Section of your manuscript: 

This research was supported by R01MH111602-05 (Dr. Thirumurthy) and T32AI055435-16A1 (Dr. Bien-Gund). 

Please note that funding information should not appear in the Acknowledgments/Funding section or other areas of your manuscript. We will only publish funding information present in the Funding Statement section of the online submission form. 

This research was supported by grants from the National Institutes of Health, R01MH111602-05 (Dr. Thirumurthy) and T32AI055435-16A1 (Dr. Bien-Gund). The funders had no role in study design, data collection and analysis, decision to publish, or preparation of the manuscript.

Reviewers' comments:

Reviewer's Responses to Questions

**Comments to the Author**

1. Is the manuscript technically sound, and do the data support the conclusions?

Reviewer #1: Yes

Reviewer #2: Yes

Reviewer #3: Yes

2. Has the statistical analysis been performed appropriately and rigorously? 

Reviewer #1: No

Reviewer #2: Yes

Reviewer #3: N/A

3. Have the authors made all data underlying the findings in their manuscript fully available?

Reviewer #1: Yes

Reviewer #2: Yes

Reviewer #3: Yes

4. Is the manuscript presented in an intelligible fashion and written in standard English?

Reviewer #1: Yes

Reviewer #2: Yes

Reviewer #3: Yes

5. Review Comments to the Author

Reviewer #1: This is an interesting analysis of an observational study nested within a previously conducted cluster randomized controlled trial of an HIV self-testing intervention in Siaya County, Kenya. The article is generally well written, however the statistical analysis is poorly described and, as a consequence, also the results of the logistic regression models are difficult to interpret.

Main points

1. My first question upon reading this was whether although the low uptake of PrEP, the intervention was associated with a reduced risk of infections in both arms of the trials. The authors acknowledge the possibility of measurement error in the exposure as PrEP use was assessed by self-report. Because of this it would be important to show that there was an association with reduced risk of infection.

2. The most critical part regards the GEE logistic regression model. First of all in Table 1 I’d report the unadjusted OR for all variables. These are important to put factors into prospective and see whether for example condom use and IPV had no association with self-reported uptake of PreP at all or whether the association could be explained by other factors and which factors.

3. Second, it is unclear which variables were actually included in the multivariable model(s) and why. In particular, authors should distinguish between these two possible scenarios:

i) Scenario 1. A single multivariable model was fitted including all the variables shown in Table 1 with the ‘behaviour’ type variables as exposures of interest and age, income and education as potential confounders. This scenario has issues as, for example, ‘condom use’ is likely to be a consequence (mediator) of ‘regular or repeated transactional sex’ and not a confounder so the results of a model mutually adjusting for these is difficult to interpret. Similarly, for example, age, income and education might be confounders for transactional sex but not for condom use, etc. In general it is expected that a different set of confounders might be suitable for each of the specific behavioural exposures.

ii) Scenario2. A separate multivariable model was constructed for each of the behavioural exposures and they were all adjusted for age, income and education (i.e. several multivariable models with 4 covariates at the time). If this was the approach uses, it should be clearly stated in the Methods and adjustments possibly revised as the latter concerns expressed for scenario i) still apply here.

4. Final point regards hypothesis testing. Authors are encouraged to calculate type 3 p-values to establish significance for categorical variables. For example, the association with relationship status is claimed to be statistically significant on the basis of the contrast specific p-value comparing ‘No relationship’ with ‘Married’. This could only be claimed if the type 3 p-value for the whole variable was <0.05.

Minor Point

Discussion page10, first paragraph. The issue is whether PreP was under-reported due to stigma or because it was not encouraged in a trial in which the primary endpoint was HIV infection.

Reviewer #2: This is an important analysis of PrEP use in cisgender women in a Kenya HIVST study. There are several areas that could be improved prior to publication.

1. Please re-define PrEP use throughout the paper. Does this mean they received a PrEP prescription at baseline? Or that they had taken PrEP prior to the baseline visit? PrEP use is very confusing in the paper and should be reviewed to define: PrEP initiation, PrEP prescription received?

2. PrEP persistence is self- reported any use (I am assuming this includes taking PrEP once?) - please can you provide more detail about # of PrEP taken in past 7 or 30 days to contextualize this better? Why was this excluded from the mv analysis? Would be helpful to understand PrEP persistence and factors associated with PrEP persistence in this cohort.

3. Did you collect data on pregnancy in the cohort? Were any pregnant women included? Can you please add this data into the analysis if so as pregnancy may have affected PrEP access/use.

4. In the methods can you please describe how PrEP was integrated into the study or clinic? How challenging was it to get a prescription? Was it integrated into their HIV testing care? Pharmacy based (or did nurses provide PrEP)

5. In the discussion please differentiate between PrEP initiation (which was very low) and persistence as "PrEP use" is very broad and there is literature to be cited for barriers to PrEP start and PrEP continuation in cisgender women that should be included.

6. Considering the above feedback, a flow chart of participants would be helpful to understand the breakdown at each study visit of PrEP start, continuation and persistence (with specific definitions).

Reviewer #3: Thanks for this manuscript. Data and experiences on routine uptake of PrEP is scarce and data like this helps to inform routine programs. The data is presented clearly and the authors state that the data is available. It is not possible to assess whether the statistical analysis has been performed appropriately and rigorously, however, the authors present a descriptive analysis and multivariate analysis that appears coherent. Some adjustments on wording should be considered.

Comments that should be observed in the manuscript:

• Consider adjusting wording related to PrEP adherence and retention in PrEP. At the moment the authors refer to “persistence on PrEP” which is not a terminology used and might lead to confusion, I would strongly consider adjusting to “retention in PrEP” as we see the authors measure here the % of women who reported use of PrEP at months 6 and 12.

• Methods section:

o could the authors elaborate more in detail how and where was PrEP offered to the women? The paper explains that women were recruited from a different clustered study from the community, and it is also stated that women were not offered actively PrEP. So, how did they get access to PrEP?

o Please, clarify on what baseline PrEP means, does it mean PrEP initiation at that time? Offered by whom? Or does it mean reported use of PrEP at the baseline visit and in that case, for how long women had been using PrEP?

o In order to understand how PrEP implementation was done it would be advisable to add a section on how the national program is implementing use of PrEP, criteria? When is it offered? And more importantly, explain programmatic information that might impact the findings, for example, what is the routing duration of PrEP refills?

o In the section “Measures” the authors should consider adding a definition on “PrEP use” as used in the manuscript, i.e. “use of PrEP at any point in time self-reported during the follow up visits”, as it needs to be clarified that it does not refer neither to uptake nor to retention in PrEP.

• Results section:

o In the point about the 1.7% of women who reported use of PrEP at baseline, can the authors clarify when those women had started PrEP, for how long they reported use of PrEP?

o The multivariable model evaluates risk factors of use of PrEP at any point of time or risk factors to report PrEP initiation. Could the authors clarify?

o In line with previous comment. From other experiences we know that retention in PrEP is very challenging and most women do not take PrEP for long time or they initiate and they don't engage on follow up visits to continue PrEP with substantial drop after the first refill. However, we still do not have enough information on why this is the case and risk factors to low retention in PrEP. Do the authors have a risk factor analysis specific of retention in PrEP?

• Discussion section:

o One point of the discussion is that low uptake of PrEP might be explained by the fact that the study evaluated PrEP uptake in the very early moments of being adopted by national policy. And they add, that there was “less active promotion of PrEP for women participating in the study compared to other pilot projects”. Can the authors elaborate on how women were aware of PrEP, how it was promoted and how they would access it?

o A point on the discussion is made on the fact that initial uptake of PrEP seem to improve overtime from baseline, to month 3 and to month 12. However, retention in PrEP seems to be stable, with an aprox 60% drop from baseline to month3 but similarly a 60% drop from first reporting use of PrEP at month3 to month12. Can authors elaborate a bit on that in the discussion? This highlights the point made on the huge challenges around retention in PrEP of which we do not know enough.

o It would be good to elaborate further on the possible factors influencing retention in PrEP, for example, what was the duration of refills of PrEP? or where PrEP is offered? We know that women at high risk of acquiring HIV like sex workers or women engaging in transactional sex have difficulties to access routine health care services; if PrEP is offered in routine health care services for a short duration of refills, this could possibly be a deterrent for continuation of PreP for women. Do the authors have any possible explanation, clarification or additional information to further include this in discussion? This would enrich the quality and added value of the paper.

o In the same line as above, the sentence in the discussion on “Interventions to support adherence and retention may borrow approaches….” could be more elaborated. While the strategies mentioned are correct, further reflection on strategies addressed to reach and access population at high risk and key populations could be made for example, peer-led approaches, longer refills of PrEP, community-based interventions for delivery of PrEP….

o In the point describing the limitations, there might be also an important bias on identifying women at high risk of acquiring HIV infection. All assessment is based on the initial questionnaire and self-reporting; typically, women engaging in transactional sex or sex workers might not disclose themselves at high risk which could have led to a potential underestimation of risk. Could the authors elaborate on that and further clarify in the discussion of limitations?

6. PLOS authors have the option to publish the peer review history of their article (what does this mean?). If published, this will include your full peer review and any attached files.

Reviewer #1: No

Reviewer #2: No

Reviewer #3: No

---

## [Author Response · Author response to Decision Letter 0]

4 Oct 2021

September 26, 2021

Dear Editor, 

Thank you for the opportunity to revise and resubmit our manuscript to PLoS One. We thank the reviewers for their comments and suggestions. The reviewers’ comments are copied and italicized below. We have itemized our responses including revisions to the manuscript, and we feel the manuscript is significantly strengthened as a result. 

 Review Comments to the Author

Reviewer #1: This is an interesting analysis of an observational study nested within a previously conducted cluster randomized controlled trial of an HIV self-testing intervention in Siaya County, Kenya. The article is generally well written, however the statistical analysis is poorly described and, as a consequence, also the results of the logistic regression models are difficult to interpret.

Main points

1. My first question upon reading this was whether although the low uptake of PrEP, the intervention was associated with a reduced risk of infections in both arms of the trials. The authors acknowledge the possibility of measurement error in the exposure as PrEP use was assessed by self-report. Because of this it would be important to show that there was an association with reduced risk of infection.

We agree that it would be important to demonstrate whether the HIV self-testing intervention had an impact on the risk of HIV acquisition, which is the primary outcome of the cluster randomized trial. The results from the trial will soon be published in The Lancet HIV, and they show that there no difference in HIV incidence between study arms. However, the intervention did increase women’s knowledge of partner HIV status as well as couples testing. Given the relatively low uptake of PrEP and the fact that HIV seroconversions were rare events, we do not believe it is possible in this study to examine the association between PrEP use and HIV infection risk. However, we think it is important and interesting to examine patterns of PrEP uptake and persistence in the study populations, and we also show that there was no significant difference in PrEP use between study arms. We have added the following sentence to the second paragraph of the Results:

“We did not observe an association between PrEP use and study arm (P=0.65).”

2. The most critical part regards the GEE logistic regression model. First of all in Table 1 I’d report the unadjusted OR for all variables. These are important to put factors into prospective and see whether for example condom use and IPV had no association with self-reported uptake of PreP at all or whether the association could be explained by other factors and which factors.

Thank you for this suggestion, and we have revised our Tables to include unadjusted associations. Table 1 now shows baseline characteristics and type III p-values (in response to this reviewer’s comment #4), and Table 2 now shows both unadjusted and adjusted odds ratios. 

3. Second, it is unclear which variables were actually included in the multivariable model(s) and why. In particular, authors should distinguish between these two possible scenarios:

i) Scenario 1. A single multivariable model was fitted including all the variables shown in Table 1 with the ‘behaviour’ type variables as exposures of interest and age, income and education as potential confounders. This scenario has issues as, for example, ‘condom use’ is likely to be a consequence (mediator) of ‘regular or repeated transactional sex’ and not a confounder so the results of a model mutually adjusting for these is difficult to interpret. Similarly, for example, age, income and education might be confounders for transactional sex but not for condom use, etc. In general it is expected that a different set of confounders might be suitable for each of the specific behavioural exposures.

ii) Scenario2. A separate multivariable model was constructed for each of the behavioural exposures and they were all adjusted for age, income and education (i.e. several multivariable models with 4 covariates at the time). If this was the approach uses, it should be clearly stated in the Methods and adjustments possibly revised as the latter concerns expressed for scenario i) still apply here.

We have clarified the Methods section regarding statistical analysis and multivariable model building. In this analysis, separate models were constructed for each of the behavioral exposures and adjusted for age, income, and education. While other model building strategies may have merits over our chosen strategy, we chose this approach due to the exploratory nature of this study and lack of literature in this field. We have added the following to the Methods section now reads:

“Multivariable models were adjusted for age (continuous), monthly income (continuous, converted to US dollars), and education (categorical), and were determined a priori by the study group. Separate multivariable models were built for each exposure and the primary outcome combination.”

4. Final point regards hypothesis testing. Authors are encouraged to calculate type 3 p-values to establish significance for categorical variables. For example, the association with relationship status is claimed to be statistically significant on the basis of the contrast specific p-value comparing ‘No relationship’ with ‘Married’. This could only be claimed if the type 3 p-value for the whole variable was <0.05.

Thank you for this suggestion. The revised Table 1 now includes type III p-values for all variables. 

Minor Point

Discussion page10, first paragraph. The issue is whether PreP was under-reported due to stigma or because it was not encouraged in a trial in which the primary endpoint was HIV infection.

Thank you for raising this point. We agree that stigma associated with PrEP may have resulted in under-reporting of PrEP use. We have added the following to the limitations section:

“Nevertheless, because this study was not conducted to evaluate PrEP, we did not expect women to over-report PrEP use. On the other hand, women may have under-reported PrEP and sexual risk behaviors due to stigma associated with HIV and with taking PrEP.”

Reviewer #2: This is an important analysis of PrEP use in cisgender women in a Kenya HIVST study. There are several areas that could be improved prior to publication.

1. Please re-define PrEP use throughout the paper. Does this mean they received a PrEP prescription at baseline? Or that they had taken PrEP prior to the baseline visit? PrEP use is very confusing in the paper and should be reviewed to define: PrEP initiation, PrEP prescription received?

Thank you for this comment. We appreciate the need to clarify our primary outcome, and agree that the language throughout the manuscript should be clarified and harmonized to reflect the emphasis on the primary outcome, PrEP uptake. The beginning of the Methods section defines PrEP uptake, and now reads:

“The primary outcome of interest was PrEP uptake, which was determined on the basis of participants’ reports of PrEP use at any of the study visits (baseline, 6, and 12 months). PrEP uptake was determined to have occurred if participants reported yes to the question: “Are you currently taking any HIV medication in order to prevent acquiring HIV (PrEP)?” Participants who reported PrEP use at any of the study visits were considered to have met the primary outcome.”

Throughout the manuscript, we have harmonized the language to indicate the primary outcome of PrEP uptake.

2. PrEP persistence is self- reported any use (I am assuming this includes taking PrEP once?) - please can you provide more detail about # of PrEP taken in past 7 or 30 days to contextualize this better? Why was this excluded from the mv analysis? Would be helpful to understand PrEP persistence and factors associated with PrEP persistence in this cohort.

We agree PrEP persistence and adherence are critical issues in implementation of this biomedical prevention tool. We have revised the “Measures” subsection of the Methods section to define PrEP persistence in our study:

“PrEP persistence was defined as self-reporting continuation of PrEP use at a follow-up visit if participants reported PrEP use at the previous visit.”

Measures of PrEP adherence that the reviewer mentions, such as the number of pills taken in the past week or month or biomarkers of adherence were not recorded as part of this study. We did not assess factors associated with PrEP persistence, due to the low numbers of women who persisted on PrEP in our cohort. We note this limitation in our Discussion section: 

“Second, we did not collect clinical or biomarker data on PrEP adherence, a key component to efficacy, and did not examine factors associated with persistence due to the small number of women reporting PrEP persistence.”

3. Did you collect data on pregnancy in the cohort? Were any pregnant women included? Can you please add this data into the analysis if so as pregnancy may have affected PrEP access/use.

Thank you for this suggestion. Pregnant women were included in this study and a measure of self-reported pregnancy was included in the survey. We have added this data into the analysis and into both Table 1 and 2. Of note, pregnancy was not associated with PrEP use in the crude or adjusted regression models; the AOR was 0.79 (95% CI 0.50, 1.25), p = 0.31. 

4. In the methods can you please describe how PrEP was integrated into the study or clinic? How challenging was it to get a prescription? Was it integrated into their HIV testing care? Pharmacy based (or did nurses provide PrEP)

In the Methods section, we describe how women were recruited from non-clinical sites. PrEP was not integrated into this study and study staff did not actively encourage or refer participants to PrEP. Since we assessed PrEP use among women in non-clinical settings, we do not collect detailed information on how PrEP was integrated into clinic settings. The beginning of the Methods section reads as follows:

“Women were recruited beginning in June 2017, one month after PrEP scale-up began in Kenya, from beach communities and “hotspots” where women engaged in transactional sex. Eligibility criteria included being HIV-negative, age ≥18 years, and reporting ≥2 sexual partners in the past 4 weeks. Individuals were excluded if they were enrolled in another HIV prevention research study… Study staff did not actively encourage or refer participants to seek PrEP at health facilities.”

5. In the discussion please differentiate between PrEP initiation (which was very low) and persistence as "PrEP use" is very broad and there is literature to be cited for barriers to PrEP start and PrEP continuation in cisgender women that should be included.

We agree that issues regarding PrEP uptake and persistence are different and focus on each issue separately in the second and third paragraphs of the Discussion, respectively. As we note in response to this Reviewer’s first and second comments, we have clarified that our primary outcome relates to PrEP uptake. We have added the following to the Discussion (paragraph 2) as it related to the low PrEP uptake, and barriers to uptake:

“The lower uptake in our study than in demonstration projects may be because there was less active promotion of PrEP for women participating in our study, compared to women reached by demonstration projects. Barriers to PrEP uptake include low risk perception, pill burden, concern of male partners, and stigma associated with HIV and PrEP (1-3).”

We describe PrEP persistence separately in the third paragraph of the Discussion, and have added to this section with additional citations. The relevant section of the third paragraph of the Discussion now reads:

“Although risk profiles among women may have changed during the study, lack of social support, changes in risk perception of HIV, pill burden, and stigma are well-described barriers to persistence on PrEP among women in Kenya (4, 5). In addition, interventions to support adherence and persistence on PrEP may borrow approaches from ART adherence strategies, such as simplified PrEP delivery models, adherence reminders, and stigma reduction activities (6, 7).”

6. Considering the above feedback, a flow chart of participants would be helpful to understand the breakdown at each study visit of PrEP start, continuation and persistence (with specific definitions).

Thank you for this comment. Figure 1 in the manuscript shows the breakdown of PrEP uptake at the study baseline, 6 months, and 12 months time points. Different shades are used to indicate the proportion of participants who persisted on PrEP at follow-up visits in 6 month intervals. A copy of this Figure is shown below:

Reviewer #3: Thanks for this manuscript. Data and experiences on routine uptake of PrEP is scarce and data like this helps to inform routine programs. The data is presented clearly and the authors state that the data is available. It is not possible to assess whether the statistical analysis has been performed appropriately and rigorously, however, the authors present a descriptive analysis and multivariate analysis that appears coherent. Some adjustments on wording should be considered.

Comments that should be observed in the manuscript:

1. Consider adjusting wording related to PrEP adherence and retention in PrEP. At the moment the authors refer to “persistence on PrEP” which is not a terminology used and might lead to confusion, I would strongly consider adjusting to “retention in PrEP” as we see the authors measure here the % of women who reported use of PrEP at months 6 and 12.

Thank you for raising this important point. The terminology of PrEP persistence has been described in multiple studies. In their editorial “Pre-exposure Prophylaxis Persistence Is a Critical Issue in PrEP Implementation” (8), Drs. Spinelli and Buchbinder highlight low rates of persistence among certain subpopulations of MSM observed in a prospective cohort study as a key gap in implementation in Clinical Infectious Diseases (9). The term PrEP persistence has also been described as an outcome in HPTN 082, recently published in PLoS Medicine (10), and in a US sample of PrEP users in Journal of the International AIDS Society (11). We have therefore elected to continue using the term “PrEP persistence” in this manuscript, although we appreciate your point that PrEP retention is an alternative term that could be used. 

To clarify in the manuscript how we defined PrEP persistence, the Measures subsection of the Methods section now reads:

“PrEP persistence was defined as self-reporting continuation of PrEP use at a follow-up visit if participants reported PrEP use at the previous visit.” 

• Methods section:

2. could the authors elaborate more in detail how and where was PrEP offered to the women? The paper explains that women were recruited from a different clustered study from the community, and it is also stated that women were not offered actively PrEP. So, how did they get access to PrEP?

Thank you for this suggestion. We have added additional details to PrEP implementation in Kenya during the early phases of the roll-out in the Methods section. It is important to note that study participants who access PrEP would have done so at local health facilities, some of which began offering PrEP during the study. The first paragraph of the Methods section now reads:

“Between June 2017 and August 2018, women were recruited from beach communities and “hotspots” where transactional sex was common. Enrollment in the study happened to begin one month after the PrEP scale-up was initiated in Kenya. During the early roll-out in Kenya, PrEP delivery efforts were focused on areas with high HIV incidence and offered in HIV clinics, antenatal clinics, and drop-in centers for key populations, including for sex workers and adolescent girls and women (2, 12). Indications for PrEP include confirmed HIV-negative status, and any of the following: a sexual partner known to be HIV positive and without an undetectable viral load, a sexual partner of unknown HIV status and at risk for HIV infection, engaging in transactional sex, history of a sexually transmitted infection, use of post-exposure prophylaxis, inconsistent condom use, and sero-discordant couples trying to conceive (13).”

3. Please, clarify on what baseline PrEP means, does it mean PrEP initiation at that time? Offered by whom? Or does it mean reported use of PrEP at the baseline visit and in that case, for how long women had been using PrEP?

We have clarified in the manuscript how we defined our primary outcome. We do not have data on how long women had been using PrEP. The Measures subsection of the Methods section now reads:

“The primary outcome of interest was self-reported current PrEP use at any of the study visits (baseline, 6, and 12 months), which we define as PrEP uptake. PrEP uptake was determined by participants who reported yes to the question: “Are you currently taking any HIV medication in order to prevent acquiring HIV (PrEP)?” Women who reported PrEP use at any of the study visits were considered to have met the primary outcome.”

4. In order to understand how PrEP implementation was done it would be advisable to add a section on how the national program is implementing use of PrEP, criteria? When is it offered? And more importantly, explain programmatic information that might impact the findings, for example, what is the routing duration of PrEP refills?

Thank you for this suggestion. We have revised the Methods section to include background on the Study Setting and PrEP scale-up in Kenya, as detailed in response to the comment #2, including criteria for PrEP in the Kenya national plan. We do not have data on PrEP refill duration. As described in the Methods section, there was heterogeneity regarding PrEP delivery models. 

5. In the section “Measures” the authors should consider adding a definition on “PrEP use” as used in the manuscript, i.e. “use of PrEP at any point in time self-reported during the follow up visits”, as it needs to be clarified that it does not refer neither to uptake nor to retention in PrEP.

Thank you for this suggestion. We have revised the definition of the primary outcome in the Measures section as noted in response to Reviewers #2 and #3. As described above, the Measures section now reads: 

“The primary outcome of interest was self-reported current PrEP use at any of the study visits (baseline, 6, and 12 months), which we define as PrEP uptake. PrEP uptake was determined by participants who reported yes to the question: “Are you currently taking any HIV medication in order to prevent acquiring HIV (PrEP)?” 

 Results section:

6. In the point about the 1.7% of women who reported use of PrEP at baseline, can the authors clarify when those women had started PrEP, for how long they reported use of PrEP?

Thank you for this question. Unfortunately, we do not know when women who reported PrEP use at baseline started PrEP. We only have information on whether they were taking PrEP at the time of the study visit.

7. The multivariable model evaluates risk factors of use of PrEP at any point of time or risk factors to report PrEP initiation. Could the authors clarify?

Thank you for this question. Risk factors were assessed at baseline, but some measures included periods up to the past 12 months. In the “Measures” subsection of the Methods, the manuscript reads: 

“Participant characteristics were assessed at baseline, including participant age, education, relationship status, primary income source, and monthly income. HIV and STI measures included self-reported STI diagnosis in the past 6 months and most recent HIV testing. Sexual risk behavior variables included having a primary partner, partner HIV status, condom use with primary partner, number of non-primary partners in the past month, condom use with non-primary partners, and regular or repeated transactional sex over the past 12 months.”

8. In line with previous comment. From other experiences we know that retention in PrEP is very challenging and most women do not take PrEP for long time or they initiate and they don't engage on follow up visits to continue PrEP with substantial drop after the first refill. However, we still do not have enough information on why this is the case and risk factors to low retention in PrEP. Do the authors have a risk factor analysis specific of retention in PrEP?

We agree that retention in care and persistence on PrEP are critically important to PrEP implementation. We observed low rates of persistence at 6- and 12-month follow-up visits. Because the numbers of individuals who persisted on PrEP were so low, we did not conduct a risk factor analysis of PrEP persistence. In the Discussion of PrEP persistence, we added the following: 

“In an implementation study of PrEP among pregnant and post-partum women, less than 40% continued PrEP after the first month, and having an HIV-positive partner was the only predictor of continuation of PrEP (2).”

Nonetheless, we agree that these are lines of future research, particularly as PrEP becomes more widespread. We have added the following to the final concluding paragraph:

“Future research can elucidate barriers and facilitators to PrEP use, adherence, and persistence among women at risk of HIV infection.” 

 Discussion section:

9. One point of the discussion is that low uptake of PrEP might be explained by the fact that the study evaluated PrEP uptake in the very early moments of being adopted by national policy. And they add, that there was “less active promotion of PrEP for women participating in the study compared to other pilot projects”. Can the authors elaborate on how women were aware of PrEP, how it was promoted and how they would access it?

Thank you for this question. We have added additional context in the Methods section to describe how women might become aware of PrEP, how it was promoted and accessed. In the Discussion section, we revised this section to describe additional barriers to both the “demand” and “supply” of PrEP during the initial roll-out. The second paragraph of the Discussion now reads:

“The lower uptake in our study than in demonstration projects may be because there was less active promotion of PrEP for women participating in our study, compared to women reached by demonstration projects. This may have diminished both demand for PrEP and PrEP availability through the public sector. Barriers to PrEP uptake include low risk perception of HIV, concern of male partners, and stigma associated with HIV and PrEP (1-3). In addition, the additional clinical resources and training associated with PrEP prescribing to an overburdened public health system may have diminished the initial impact of the PrEP roll-out (7).”

10. A point on the discussion is made on the fact that initial uptake of PrEP seem to improve overtime from baseline, to month 3 and to month 12. However, retention in PrEP seems to be stable, with an aprox 60% drop from baseline to month3 but similarly a 60% drop from first reporting use of PrEP at month3 to month12. Can authors elaborate a bit on that in the discussion? This highlights the point made on the huge challenges around retention in PrEP of which we do not know enough.

We agree that high rates of PrEP discontinuation are a major challenge to implementation. We have added to the discussion on both “demand” (patient-level) and “supply” (provider and systems-level) barriers to continuation on PrEP. The third paragraph of the Discussion section now reads: 

“Consistent with previously published data, we observed low persistence on PrEP, with less than 50% of women reporting persistence on PrEP at 6-month follow-up visits (5, 14, 15). Suboptimal persistence on PrEP is a pervasive challenge to successful PrEP implementation. In an implementation study of PrEP among pregnant and post-partum women, less than 40% continued PrEP after the first month, and having an HIV-positive partner was the only predictor of continuation of PrEP (2). Although risk profiles among women may have changed during the study, changes in risk perception of HIV, pill burden, lack of social support, and stigma have been described as barriers to persistence on PrEP among women in Kenya (4, 5). From a PrEP provider perspective, providers in public health clinics have cited potential strategies such as reducing the frequency of clinic visits and longer refill dates, as well as sending clinic reminders to support persistence on PrEP (7). Interventions to support adherence and persistence on PrEP may borrow approaches from ART adherence strategies, such as simplified PrEP delivery models (including pharmacy-delivered or other non-clinical sites), adherence clubs, and stigma reduction activities (6, 7).”

11. It would be good to elaborate further on the possible factors influencing retention in PrEP, for example, what was the duration of refills of PrEP? or where PrEP is offered? We know that women at high risk of acquiring HIV like sex workers or women engaging in transactional sex have difficulties to access routine health care services; if PrEP is offered in routine health care services for a short duration of refills, this could possibly be a deterrent for continuation of PreP for women. Do the authors have any possible explanation, clarification or additional information to further include this in discussion? This would enrich the quality and added value of the paper.

We agree that provider- and systems-level barriers need to be addressed to support continuation on PrEP. As we have described in our response to comment #10 above, we have now revised this section of the Discussion to include potential strategies to reduce these barriers to PrEP continuation. 

12. In the same line as above, the sentence in the discussion on “Interventions to support adherence and retention may borrow approaches….” could be more elaborated. While the strategies mentioned are correct, further reflection on strategies addressed to reach and access population at high risk and key populations could be made for example, peer-led approaches, longer refills of PrEP, community-based interventions for delivery of PrEP….

Thank you for this comment. We have elaborated on potential “supply” strategies to increase the delivery of PrEP, as described in our response to comment #10 above. 

13. In the point describing the limitations, there might be also an important bias on identifying women at high risk of acquiring HIV infection. All assessment is based on the initial questionnaire and self-reporting; typically, women engaging in transactional sex or sex workers might not disclose themselves at high risk which could have led to a potential underestimation of risk. Could the authors elaborate on that and further clarify in the discussion of limitations?

We agree that any self-reported measures may be under-reported, particularly regarding stigmatized topics such as transactional sex and sex work. We have added this as a potential limitation, in addition to the point Reviewer #1 commented on, that PrEP may also have been underreported due to stigma. Regarding the potential for bias, if under-reporting of sexual risk and PrEP are both mediated by stigma, we would not expect there to be a significant bias affecting study results. To address this point, we have added the following to the Limitations section: 

“On the other hand, women may have under-reported PrEP and sexual risk behaviors due to stigma associated with taking HIV and PrEP.”

We look forward to hearing from you regarding our revisions and responses and hope that the edited manuscript is now acceptable for publication in PLoS One.

Sincerely,

Cedric Bien-Gund, MD

3400 Spruce Street

3 Silverstein, Suite E

Division of Infectious Diseases

Perelman School of Medicine, University of Pennsylvania

Philadelphia, Pennsylvania

Cedric.bien-gund@pennmedicine.upenn.edu

Tele: 215-615-4724

Fax: 215-662-7611

 

References cited in reply letter: 

1. Pintye J, Beima-Sofie KM, Makabong'O PA, Njoroge A, Trinidad SB, Heffron RA, et al. HIV-Uninfected Kenyan Adolescent and Young Women Share Perspectives on Using Pre-Exposure Prophylaxis During Pregnancy. AIDS Patient Care and STDs. 2018;32(12):538-44.

2. Kinuthia J, Pintye J, Abuna F, Mugwanya KK, Lagat H, Onyango D, et al. Pre-exposure prophylaxis uptake and early continuation among pregnant and post-partum women within maternal and child health clinics in Kenya: results from an implementation programme. The lancet HIV. 2020;7(1):e38-e48.

3. Celum CL, Delany-Moretlwe S, Baeten JM, van der Straten A, Hosek S, Bukusi EA, et al. HIV pre-exposure prophylaxis for adolescent girls and young women in Africa: from efficacy trials to delivery. Journal of the International AIDS Society. 2019;22 Suppl 4(Suppl Suppl 4):e25298.

4. Kyongo J, Kiragu M, Karuga R, Ochieng C, Ngunjiri A, Wachihi C, et al. How long will they take it? Oral pre-exposure prophylaxis (PrEP) retention for female sex workers, men who have sex with men and young women in a demonstration project in Kenya. 22nd International AIDS Conference; Amsterdam, Netherlands2018.

5. Mugwanya KK, Pintye J, Kinuthia J, Abuna F, Lagat H, Begnel ER, et al. Integrating preexposure prophylaxis delivery in routine family planning clinics: A feasibility programmatic evaluation in Kenya. PLOS Medicine. 2019;16(9):e1002885.

6. Irungu EM, Baeten JM. PrEP rollout in Africa: status and opportunity. Nature medicine. 2020;26(5):655-64.

7. Irungu EM, Odoyo J, Wamoni E, Bukusi EA, Mugo NR, Ngure K, et al. Process evaluation of PrEP implementation in Kenya: adaptation of practices and contextual modifications in public HIV care clinics. J Int AIDS Soc. 2021;24(9):e25799.

8. Spinelli MA, Buchbinder SP. Pre-exposure Prophylaxis Persistence Is a Critical Issue in PrEP Implementation. Clinical Infectious Diseases. 2019;71(3):583-5.

9. Serota DP, Rosenberg ES, Sullivan PS, Thorne AL, Rolle C-PM, Del Rio C, et al. Pre-exposure Prophylaxis Uptake and Discontinuation Among Young Black Men Who Have Sex With Men in Atlanta, Georgia: A Prospective Cohort Study. Clinical Infectious Diseases. 2019;71(3):574-82.

10. Celum C, Hosek S, Tsholwana M, Kassim S, Mukaka S, Dye BJ, et al. PrEP uptake, persistence, adherence, and effect of retrospective drug level feedback on PrEP adherence among young women in southern Africa: Results from HPTN 082, a randomized controlled trial. PLOS Medicine. 2021;18(6):e1003670.

11. Coy KC, Hazen RJ, Kirkham HS, Delpino A, Siegler AJ. Persistence on HIV preexposure prophylaxis medication over a 2-year period among a national sample of 7148 PrEP users, United States, 2015 to 2017. J Int AIDS Soc. 2019;22(2):e25252.

12. Masyuko S, Mukui I, Njathi O, Kimani M, Oluoch P, Wamicwe J, et al. Pre-exposure prophylaxis rollout in a national public sector program: the Kenyan case study. Sexual health. 2018;15(6):578-86.

13. National AIDS and STI Control Programme (NASCOP). Preliminary KENPHIA 2018 Report. Nairobi 2018.

14. Koss CA, Charlebois ED, Ayieko J, Kwarisiima D, Kabami J, Balzer LB, et al. Uptake, engagement, and adherence to pre-exposure prophylaxis offered after population HIV testing in rural Kenya and Uganda: 72-week interim analysis of observational data from the SEARCH study. The Lancet HIV. 2020;7(4):e249-e61.

15. Were D, Musau A, Mutegi J, Ongwen P, Manguro G, Kamau M, et al. Using a HIV prevention cascade for identifying missed opportunities in PrEP delivery in Kenya: results from a programmatic surveillance study. Journal of the International AIDS Society. 2020;23 Suppl 3(Suppl 3):e25537.

---

## [Decision Letter · Decision Letter 1]

9 Dec 2021

PONE-D-21-19374R1Adoption of HIV pre-exposure prophylaxis among women at high risk of HIV infection in KenyaPLOS ONE

Dear Dr. Bien-Gund,

Thank you for submitting your manuscript to PLOS ONE. After careful consideration, we feel that it has merit but does not fully meet PLOS ONE’s publication criteria as it currently stands. Therefore, we invite you to submit a revised version of the manuscript that addresses the points raised during the review process.

We look forward to receiving your revised manuscript.

Kind regards,

Joseph KB Matovu, Ph.D.

Academic Editor

PLOS ONE

Journal Requirements:

Reviewers' comments:

Reviewer's Responses to Questions

**Comments to the Author**

1. If the authors have adequately addressed your comments raised in a previous round of review and you feel that this manuscript is now acceptable for publication, you may indicate that here to bypass the “Comments to the Author” section, enter your conflict of interest statement in the “Confidential to Editor” section, and submit your "Accept" recommendation.

Reviewer #1: (No Response)

Reviewer #2: All comments have been addressed

Reviewer #3: All comments have been addressed

2. Is the manuscript technically sound, and do the data support the conclusions?

Reviewer #1: Partly

Reviewer #2: Yes

Reviewer #3: Yes

3. Has the statistical analysis been performed appropriately and rigorously? 

Reviewer #1: No

Reviewer #2: Yes

Reviewer #3: I Don't Know

4. Have the authors made all data underlying the findings in their manuscript fully available?

Reviewer #1: Yes

Reviewer #2: Yes

Reviewer #3: (No Response)

5. Is the manuscript presented in an intelligible fashion and written in standard English?

Reviewer #1: Yes

Reviewer #2: Yes

Reviewer #3: Yes

6. Review Comments to the Author

Reviewer #1: The authors did a good job at addressing my previous concerns.

However I have additional comments to this revised version

Point#2

The authors claim that separate models were constructed for each of the behavioural exposures which were adjusted for age, income, and education. Nevertheless the results of these models are reported in a single table (Table 2). This is confusing because ORs are not all interpretable in the same way [Westreich D, Greenland S. The table 2 fallacy: presenting and interpreting confounder and modifier coefficients. Am J Epidemiol. 2013 Feb 15;177(4):292-8.] I recommend splitting the table in panels including the results from these separate models

Point#4

Univariable type 3 p-values have been added in Table 1. The request was referring to the addition of the type III p-values from the multivariable models in Table 2.

I'd also recommend that the authors tailor the conclusions regarding the evidence to reject the null hypothesis of no association using these type 3 p-values and modify Results and Discussion accordingly

Reviewer #2: Thank you for the revisions and edits. All of my edits/suggestions have been addressed. No further comments.

Reviewer #3: (No Response)

7. PLOS authors have the option to publish the peer review history of their article (what does this mean?). If published, this will include your full peer review and any attached files.

Reviewer #1: No

Reviewer #2: No

Reviewer #3: No

---

## [Author Response · Author response to Decision Letter 1]

4 Apr 2022

March 15, 2022

Dear PLoS One Editor,

Thank you for the opportunity revise our manuscript to PLoS One. We thank the reviewers for their positive response to our revision. We have responded to the remaining comments below. The reviewers’ comments are copied and italicized.

Reviewer #1: The authors did a good job at addressing my previous concerns.

However I have additional comments to this revised version

1) The authors claim that separate models were constructed for each of the behavioural exposures which were adjusted for age, income, and education. Nevertheless the results of these models are reported in a single table (Table 2). This is confusing because ORs are not all interpretable in the same way [Westreich D, Greenland S. The table 2 fallacy: presenting and interpreting confounder and modifier coefficients. Am J Epidemiol. 2013 Feb 15;177(4):292-8.] I recommend splitting the table in panels including the results from these separate models

Thank you for this comment. We agree that the results in Table 2 could be clarified to indicate that the results are from separate models, and not from a same model. We have added the following legend to the bottom of Table 2 to clarify this point: “Separate multivariable models for each variable were constructed adjusting for age, education, and income.” Table 2 thus shows the results of multiple, separate models, as the reviewer recommends. 

2) Univariable type 3 p-values have been added in Table 1. The request was referring to the addition of the type III p-values from the multivariable models in Table 2.

I'd also recommend that the authors tailor the conclusions regarding the evidence to reject the null hypothesis of no association using these type 3 p-values and modify Results and Discussion accordingly

We have revised Table 2 to include type III p-values in the univariable and multivariable models in Table 2. Using these type III p-values did not substantively change the results or conclusions of our study. 

We look forward to hearing from you regarding our revisions and responses and hope that the edited manuscript is now acceptable for publication in PLoS One.

Sincerely,

Cedric Bien-Gund, MD

3400 Spruce Street

3 Silverstein, Suite E

Division of Infectious Diseases

Perelman School of Medicine, University of Pennsylvania

Philadelphia, Pennsylvania

Cedric.bien-gund@pennmedicine.upenn.edu

Tele: 215-615-4724

Fax: 215-662-7611

---

## [Decision Letter · Decision Letter 2]

13 Jun 2022

PONE-D-21-19374R2Adoption of HIV pre-exposure prophylaxis among women at high risk of HIV infection in KenyaPLOS ONE

Dear Dr. Bien-Gund,

Thank you for submitting your manuscript to PLOS ONE. After careful consideration, we feel that it has merit but does not fully meet PLOS ONE’s publication criteria as it currently stands. Therefore, we invite you to submit a revised version of the manuscript that addresses the points raised during the review process.

We look forward to receiving your revised manuscript.

Kind regards,

Joseph KB Matovu, Ph.D.

Academic Editor

PLOS ONE

Journal Requirements:

Reviewers' comments:

Reviewer's Responses to Questions

**Comments to the Author**

1. If the authors have adequately addressed your comments raised in a previous round of review and you feel that this manuscript is now acceptable for publication, you may indicate that here to bypass the “Comments to the Author” section, enter your conflict of interest statement in the “Confidential to Editor” section, and submit your "Accept" recommendation.

Reviewer #3: (No Response)

2. Is the manuscript technically sound, and do the data support the conclusions?

Reviewer #3: Yes

3. Has the statistical analysis been performed appropriately and rigorously? 

Reviewer #3: N/A

4. Have the authors made all data underlying the findings in their manuscript fully available?

Reviewer #3: (No Response)

5. Is the manuscript presented in an intelligible fashion and written in standard English?

Reviewer #3: Yes

6. Review Comments to the Author

Reviewer #3: Thank you for the revised manuscript and substantial improvement.

The only comment in this review refers to how the uptake data is presented and I would like to request a small clarification. In Page 7, 2nd paragraph: "A total of 138 women reported PrEP uptake in first year". However, in the stratification of PrEP uptake at baseline, 6-months and 12-months it seems that the uptake at 6-months and 12-months is presented together with persistence (while it refers only to uptake). Or the data is not adding up: 35+61+87=183. Can the authors confirm the absolute numbers of the uptake data and the uptake in the different observation moments? Fig 1 shows PrEP uptake and persistence. In the pragraph I would suggest to focus on presenting only uptake (reported new initiations in total and in the different time-points (as per objectives) to have the data overview of PrEP uptake.

For the rest, no additional comments.

Best regards,

7. PLOS authors have the option to publish the peer review history of their article (what does this mean?). If published, this will include your full peer review and any attached files.

Reviewer #3: No

---

## [Author Response · Author response to Decision Letter 2]

30 Jun 2022

June 15, 2022

We thank the reviewers for their positive review and suggestion (in quotes), which we have responded to below. 

"The only comment in this review refers to how the uptake data is presented and I would like to request a small clarification. In Page 7, 2nd paragraph: "A total of 138 women reported PrEP uptake in first year". However, in the stratification of PrEP uptake at baseline, 6-months and 12-months it seems that the uptake at 6-months and 12-months is presented together with persistence (while it refers only to uptake). Or the data is not adding up: 35+61+87=183. Can the authors confirm the absolute numbers of the uptake data and the uptake in the different observation moments? Fig 1 shows PrEP uptake and persistence. In the paragraph I would suggest to focus on presenting only uptake (reported new initiations in total and in the different time-points (as per objectives) to have the data overview of PrEP uptake."

Thank you for raising this important point of clarification. We agree that this paragraph is confusing and have taken the reviewer’s suggestion to focus on new PrEP initiations, while discussing PrEP persistence separately. As the reviewer correctly identifies, the previous version of the manuscript reported totals that included individuals who have initiated PrEP as well as individuals who have persisted on PrEP. We now focus on reporting new PrEP initiations. The beginning of the second paragraph on Page 7 now reads: 

“A total of 138 (6.6%) women reported PrEP uptake during the first year of the study. At baseline, 1.7% (35/2,086) reported PrEP uptake. At the 6-month visit, an additional 46 women reported new PrEP initiation, and at the 12-month visit, an additional 57 women reported new PrEP initiation (Figure 1).” 

We look forward to hearing from you regarding our revisions and responses and hope that the edited manuscript is now acceptable for publication in PLoS One.

Sincerely,

Cedric Bien-Gund, MD

Perelman School of Medicine, University of Pennsylvania

Cedric.bien-gund@pennmedicine.upenn.edu

---

## [Editor Report · Decision Letter 3]

9 Aug 2022

Adoption of HIV pre-exposure prophylaxis among women at high risk of HIV infection in Kenya

PONE-D-21-19374R3

Dear Dr. Bien-Gund,

We’re pleased to inform you that your manuscript has been judged scientifically suitable for publication and will be formally accepted for publication once it meets all outstanding technical requirements.

Kind regards,

Joseph KB Matovu, Ph.D.

Academic Editor

PLOS ONE
---

## [Editor Report · Acceptance letter]

31 Aug 2022

PONE-D-21-19374R3 

Adoption of HIV pre-exposure prophylaxis among women at high risk of HIV infection in Kenya 

Dear Dr. Bien-Gund:

I'm pleased to inform you that your manuscript has been deemed suitable for publication in PLOS ONE. Congratulations! Your manuscript is now with our production department. 

Kind regards, 

on behalf of

Dr. Joseph KB Matovu 

Academic Editor

PLOS ONE